# A global perspective of advanced practice nursing research: A review of systematic reviews

Kelley Kilpatrick[1,2]*, Isabelle Savard[3], Li-Anne Audet[3], Gina Costanzo[3], Mariam Khan[3], Renée Atallah[3], Mira Jabbour[2], Wentao Zhou[4,5], Kathy Wheeler[6], Elissa Ladd[7], Deborah C. Gray[8], Colette Henderson[9], Lori A. Spies[10], Heather McGrath[11], Melanie Rogers[12]

1 Susan E. French Chair in Nursing Research and Innovative Practice, Ingram School of Nursing, Faculty of Medicine and Health Sciences, McGill University, Montréal, Québec, Canada, 2 Centre Intégré Universitaire de Santé et de Services sociaux de l'Est-de-l'Île-de-Montréal (CIUSSS-EMTL), Montréal, Québec, Canada, 3 Ingram School of Nursing, Faculty of Medicine and Health Sciences, McGill University, Montréal, Québec, Canada, 4 Alice Lee Centre for Nursing Studies, Yong Loo Lin School of Medicine, National University of Singapore; National University Health System, Singapore, Singapore, 5 Singapore National Neuroscience Institute, Singapore, Singapore, 6 College of Nursing, University of Kentucky, Lexington, Kentucky, United States of America, 7 MGH Institute of Health Professions, School of Nursing, Boston, Massachusetts, United States of America, 8 School of Nursing, Old Dominion University, Virginia Beach, Virginia, United States of America, 9 School of Health Sciences, University of Dundee, Dundee, Scotland, United States of America, 10 Louise Herrington School of Nursing, Baylor University, Dallas, Texas, United States of America, 11 St James Public Health Services, Montego Bay, St James, Jamaica, 12 Department of Nursing and Midwifery, University of Huddersfield, Queensgate, Huddersfield, United Kingdom

☯ These authors contributed equally to this work.
* kelley.kilpatrick@mcgill.ca

**Data Availability Statement:** All relevant data are within the manuscript and its Supporting Information files.

## Abstract

### Introduction

The World Health Organization (WHO) called for the expansion of all nursing roles, including advanced practice nurses (APNs), nurse practitioners (NPs) and clinical nurse specialists (CNSs). A clearer understanding of the impact of these roles will inform global priorities for advanced practice nursing education, research, and policy.

### Objective

To identify gaps in advanced practice nursing research globally.

### Materials and methods

A review of systematic reviews was conducted. We searched CINAHL, Embase, Global Health, Healthstar, PubMed, Medline, Cochrane Library, DARE, Joanna Briggs Institute EBP, and Web of Science from January 2011 onwards, with no restrictions on jurisdiction or language. Grey literature and hand searches of reference lists were undertaken. Review quality was assessed using the Critical Appraisal Skills Program (CASP). Study selection, data extraction and CASP assessments were done independently by two reviewers. We

**Funding:** This work is supported by the McGill University Faculty of Medicine and Health Sciences and the Newton Foundation via the Susan E. French Chair in Nursing Research and Innovative Practice held by KK. KK is also supported by a Fonds de recherche du Québec-Santé (https://frq.gouv.qc.ca/en/health/) Research Scholar Senior (Award Number 298573) salary award. There was no additional external funding received for this study, and the authors received no specific funding for this work. All the funders had no role in study design, data collection and analysis, decision to publish, or preparation of the manuscript.

**Competing interests:** The authors have declared that no competing interests exist.

extracted study characteristics, country and outcome data. Data were summarized using narrative synthesis.

## Results

We screened 5840 articles and retained 117 systematic reviews, representing 38 countries. Most CASP criteria were met. However, study selection by two reviewers was done inconsistently and language and geographical restrictions were applied. We found highly consistent evidence that APN, NP and CNS care was equal or superior to the comparator (e.g., physicians) for 29 indicator categories across a wide range of clinical settings, patient populations and acuity levels. Mixed findings were noted for quality of life, consultations, costs, emergency room visits, and health care service delivery where some studies favoured the control groups. No indicator consistently favoured the control group. There is emerging research related to Artificial Intelligence (AI).

## Conclusion

There is a large body of advanced practice nursing research globally, but several WHO regions are underrepresented. Identified research gaps include AI, interprofessional team functioning, workload, and patients and families as partners in healthcare.

### PROSPERO registration number

CRD42021278532.

## Introduction

Globally, nurses make up approximately half of the healthcare workforce [1]. The World Health Organization (WHO) has called for the broadening of nurses' scope of practice to promote equitable care delivery and respond to the growing demands for healthcare services [1, 2]. According to the International Council of Nurses (ICN), nurses in advanced practice roles have a minimum of Master's-level preparation in addition to in-depth clinical expertise and complex decision-making skills [3]. Globally, diverse titles and varying nomenclature are used for these roles, with the most widely used being the titles advanced practice nurses (APNs), nurse practitioners (NPs) and clinical nurse specialists (CNSs) [3]. Advanced practice nurses ensure direct care to patients and families with acute, chronic or complex conditions [3, 4]. In addition to providing direct care, APNs, NPs and CNSs support healthcare teams to deliver patient care and improve access to services [5–7]. Advanced practice nursing roles are growing exponentially worldwide [8]. However, an understanding of the impact of these roles has been uneven across the globe because of inconsistent definitions of roles and titles, and a lack of role clarity. A recent umbrella review of primary healthcare NPs [9] documented that an average of three countries per review were found across the 44 systematic reviews. Although there are systematic reviews of APNs and CNSs in other clinical settings (e.g., Allsop et al., 2021; Audet et al., 2021; Kerr et al. 2021) [6, 7, 10], no synthesis of this body of evidence is available using recognized advanced practice role definitions, making it difficult to compare advanced practice nursing roles internationally.

It is challenging to distinguish between the roles of nurses and advanced practice nurses [11]. For example, some jurisdictions (e.g., Belgium) employ the term nurse-led for all nursing

roles [1, 8, 12, 13]. A clearer understanding of the roles that are in place, titles used across the globe and relevant outcomes will support optimal use of advanced practice nursing roles, and inform global priorities for advanced practice nursing education, research, and policy reform.

To identify current gaps in advanced practice nursing research globally, we conducted a review of systematic reviews of studies examining APNs, NPs or CNSs using recognized advanced practice nursing role definitions [3, 4]. The overview sought to answer the following question: Do current systematic reviews that include APNs, NPs or CNSs represent countries where these roles are found globally? The three following objectives were pursued:

1. Identify the countries included in systematic reviews of APNs, NPs or CNSs;

2. Describe the types of included studies, study population, role definitions, and settings identified in the systematic reviews; and

3. Examine the types of outcomes of APN, NP or CNS roles included in systematic reviews globally.

## Materials and methods

The protocol for this overview was developed a priori and published (PROSPERO International Prospective Register of Systematic Reviews CRD42021278532) [14–16]. In essence, the review of systematic reviews describes the literature related to advanced practice nursing globally to identify key research gaps. The methods used in an umbrella review of indicators sensitive to primary healthcare NP practice formed the basis for the current review [9, 17]. The descriptive table, critical appraisal and outcomes table are presented in the appendices given the large size of these tables.

### Inclusion criteria

**Types of studies.**  Published and unpublished systematic reviews identified between January 1st 2011 and April 3rd 2023 were examined. No restrictions on jurisdiction or language were applied. The review had to 1) provide elements of a research question (i.e., PICOS) related to advanced practice nursing; and 2) detail inclusion and exclusion criteria to be considered [18]. Systematic reviews were included if the advanced practice nursing role was clearly defined and the APN, NP or CNS had decision-making autonomy [3, 4].

**Types of participants.**  Participants included individual, groups or communities. Patients were from all age groups or health conditions receiving APN, NP or CNS care regardless of the type (e.g., long-term care), size, and location (e.g., urban/rural) of the facility. All members of the healthcare team across all types, sizes, and locations of facilities were included (e.g., physicians, nurses).

**Types of interventions.**  Studies of APNs, NPs or CNSs in any sector were considered to capture representative country-level data where the roles are implemented. We identified studies in acute care and primary care settings. Acute care was defined as in-hospital or specialized ambulatory care to address specific health conditions [19]. Primary care referred to the entry point of the healthcare system where patients receive comprehensive healthcare services for common health concerns [20].

Advanced practice nursing role dimensions include clinical and non-clinical activities related to education, research, and administration/leadership [21, 22]. APNs are nurses who have acquired in-depth expertise, complex decision-making skills and advanced clinical competencies [3, 4]. Consistent with ICN definitions, master's or doctoral educational preparation is recommended and in many countries is required with national board certification for

licensure and entry-level practice [3, 4]. Given the diversity of terms used globally to identify APNs, NPs, and CNSs, research team members were actively involved in determining if role titles specific to their region represented an advanced practice nursing role (e.g., nurse-led).

NPs are autonomous clinicians who practice in acute and primary care (including long-term care (LTC), home care). NPs assess, diagnose, treat, and manage acute episodic and chronic illnesses. They order and interpret diagnostic and laboratory tests, prescribe medication and non-pharmacologic therapies. NPs are health promotion and illness prevention experts and as such teach and counsel individuals and communities. In addition to clinical practice, they are clinical and health care system leaders, interdisciplinary consultants, and patient advocates [3, 5].

CNS roles include practice, consultation, education, research and leadership [3, 4]. CNSs have in-depth expertise in a nursing specialty and help solve complex healthcare issues. They are leaders in the development of clinical practice guidelines, promoting evidence-informed practice, and facilitating system change [3, 4]. CNS specialty-areas of practice may be with specific patient populations (frail elderly), settings (ICU), disease (cancer care), or type of care (post-operative care).

**Types of comparators.** We extracted data related to the comparator (i.e., control). Comparator groups included usual care, care provided by other healthcare professionals (e.g., physicians), and adherence to best practice guidelines.

**Types of outcomes.** Outcomes were categorized as patient (e.g., quality of life), provider (e.g., education), and health system (e.g., costs, scope of practice) levels.

## Exclusion criteria

We excluded reviews addressing broad research questions (e.g., scoping reviews) and reviews related to physician assistants, certified registered nurse anesthetists, and nurse midwives. We also excluded reviews where the impact of the APNs, NPs or CNSs could not be teased out or was not reported separately from other provider roles. A list of excluded reviews are provided in S1 Appendix.

## Database search

We searched the following electronic databases from January 1st 2011 to April 3rd 2023: Cumulative Index to Nursing and Allied Health Literature (CINAHL), Embase, Global Health, Healthstar, PubMed, Medline, Cochrane Library Database of Systematic Reviews and Controlled Trials Register, Database of Abstracts of Reviews of Effects (DARE), Joanna Briggs Institute (JBI) EBP, and Web of Science [16]. This time period was selected to capture the most recent trends in advanced practice nursing research and because evidence in approximately half of reviews is outdated five years after publication [16]. The keywords and search strategies are available in S2 Appendix. Search filters based on the CADTH systematic reviews and meta-analyses search and one developed by Lunny et al. for reviews of systematic reviews were used [23, 24]. The full preliminary search strategy developed for the PubMed database was subsequently adapted to each electronic database (See S2 Appendix). We adapted strategies reviewed by an academic librarian that have been used successfully in previous reviews [9, 25]. In addition, hand searches of reference lists of all relevant reviews were conducted to identify additional studies.

The grey literature was searched for the same period using the following websites and tools: World Health Organization, Organization for Economic Co-operation and Development (OECD), CADTH Information Services, CADTH Grey Matters Tool, International Council of Nurses (ICN), and ProQuest Dissertation and Theses. We searched the PROSPERO

International Prospective Register of Systematic Reviews to identify registered review protocols. For each website, the content was searched using the same search terms as those used for the published literature. If there was not an inherent search function on the website, a search was completed of all webpages and weblinks. The search strategy for the grey literature can be found in S3 Appendix.

## Study selection

All reviewers were trained to use the screening instrument and inclusion/exclusion criteria to enhance inter-rater agreement [16]. Three training sessions were conducted. The retained studies were uploaded into the EndNote and RAYYAN software, and duplicates were removed [26]. Two reviewers independently screened titles and abstracts using the predefined inclusion/exclusion criteria, and recommended exclusion or further full-text review. Inter-rater agreement was estimated using Cohen's kappa (κ). Moderate agreement was found (κ = 0.456) with a percentage of agreement equal to 81.2 [27].

To be retained, papers had to be identified as a systematic review, and focus on an advanced practice nursing role or intervention. Full text reviews were conducted if no abstract was available or if it contained insufficient information. Any discrepancies were discussed among the reviewers until consensus was reached. A third reviewer to function as a tie-breaker was available but consensus was reached on all reviews and the additional review was not required.

## Data extraction

Six training sessions were conducted with reviewers over a 6-month period to review data extractions tools and answer questions. Individualized feedback was provided by the primary author for two practice articles per reviewer. Data were extracted by one reviewer and checked independently by a second reviewer. A third reviewer revised all the papers for consistency. Discrepancies were resolved by consensus. A structured tool developed for a previous overview was adapted and pilot-tested by the research team [17, 25]. Descriptive characteristics included: first author and publication year; review aims; number of electronic databases searched; countries where studies were conducted; number and types of studies included in the review; APN role and population; intervention and comparator; synthesis methods, funding source, APN, NP or CNS involvement in the conduct of the review. Outcomes related to patient, provider, health system, education, policy, and scope of practice were extracted for each paper [28]. Results were categorized as equal to statistically significant in favour of the intervention or control group.

**Design of included studies.**   The addition or expanded use of APNs, NPs and CNSs represents a complex healthcare system intervention [29]. Our overview considered studies that employed diverse research methods including randomized and non-randomized studies and qualitative methodologies.

## Assessment of review quality

To enhance inter-rater agreement, two training sessions with three practice articles (poor to high quality) were conducted with reviewers to examine the Critical Appraisal Skills Program (CASP) tool. Two reviewers independently assessed each review's methodologic quality using CASP [30] systematic review checklist. The instrument includes 10 items. To help standardize our assessments, we considered if the review addressed a clear question, if the protocol had been developed a priori, if elements of a PICO question could be found, or if the review aims were clearly stated. For syntheses of qualitative studies, we determined if the qualitative approach was identified and the methods appropriate. Inter-rater agreement was assessed

using Cohen's kappa. We considered that the 'can't tell' response was equivalent to the item not being met. No study was excluded based on methodologic quality because our overview aimed to capture the countries where advanced practice nursing research had been undertaken. Disagreements were reviewed and discussed until the reviewers reached a consensus. A summary of the CASP ratings can be found in S1 Table.

## Data synthesis

A narrative synthesis was developed using an iterative process [31]. Summary tables outlined key review characteristics (e.g., countries where studies were conducted), type of advanced practice nursing role, outcomes, and quality assessment. We kept a record of all review-related decisions. Additional quantitative analysis was not conducted on combined reviews to avoid the potential risk of including studies that appear in more than one review [32].

## Results

We screened 5840 articles after removing duplicates (Fig 1). Of these, full text review was completed on 961 studies. We retained 117 systematic reviews in our overview representing 1653 primary studies. The reviews were published in 121 papers between 2011 and 2023 with six reviews published as constellation papers. Each review included an average of 14.1 papers (Standard deviation (SD): 13.2, range: 1 to 79). Articles were cited once (n = 1193, 86%), twice (n = 137, 10%), three times or more (n = 51, 4%). Reviews were published in English, Farsi, German, Mandarin, and Spanish languages. APNs, NPs or CNSs were included as a co-author in 37% of reviews. The funding source was indicated in 36%.

Positive assessments of the CASP items ranged from 27% to 99%. Inter-rater agreement was almost perfect with Cohen's kappa ranging from 91% to total agreement for items in section C related to local application of results (S1 Table). Reviews were downgraded for the criterion related to study identification primarily because review authors did not clearly state that: 1) two reviewers independently screened titles and abstracts; 2) two reviewers checked the extractions; and 3) language or country restrictions were applied.

*To address Aims 1 and 2*, study characteristics are presented in S2 Table. Overall, 38 countries were identified with an average of 3.4 countries (SD: 2.5, range: 1 to 13) per review. Twelve reviews did not report on the included countries. The number of systematic reviews per country is reported in Fig 2 with the United States and United Kingdom reporting the largest number of reviews. Fig 3 represents the advanced practice nursing roles and countries where the roles are represented.

*To address Aim 3*, review outcomes were categorized into broader categories at the patient (S3 Table), provider (S4 Table) and health system levels (S5 Table). The emergence of Artificial Intelligence (AI) and advanced practice nursing roles was documented separately given the transversal role that AI can play across levels (S6 Table). Patient indicators include activities of daily living (ADLs), adaptation to health conditions, clinical outcomes, diagnosis, education-patient, mortality, morbidity, patient adherence, quality of life, satisfaction, and signs and symptoms. Provider indicator categories include adherence to best practice guidelines-provider, education-provider, illness prevention, interprofessional team functioning, prescribing, and satisfaction-provider. Health system indicator categories include access to care, consultations, costs, emergency room visits, health care service delivery, hospitalization, length of stay, patient safety, quality of care, scope of practice, and wait times. Artificial intelligence (AI)-Health technology was added as a cross cutting indicator category given its broad application. Each category is summarized below.

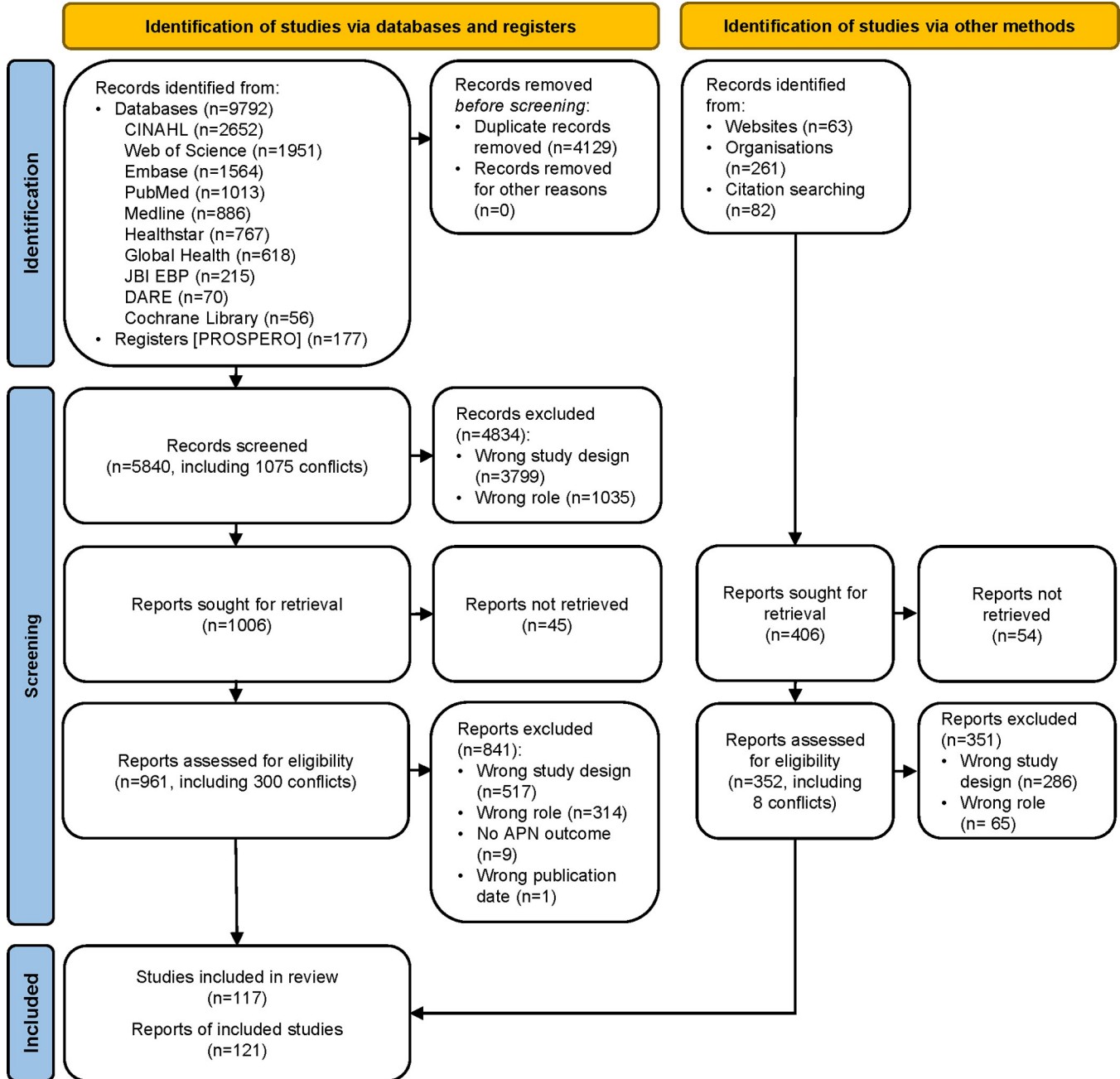

**Fig 1. PRISMA 2020 flow diagram.** * * *From*: Page MJ, McKenzie JE, Bossuyt PM, Boutron I, Hoffmann TC, Mulrow CD, et al. The PRISMA 2020 statement: an updated guideline for reporting systematic reviews. BMJ. 2021;372:n71. doi: 10.1136/bmj.n71. For more information, visit: http://www.prisma-statement. org/.

## Patient indicators

**Activities of Daily Living (ADLs)** were examined in nine reviews [7, 10, 33–40]. Reviews included APNs in acute and primary care and NPs primary care. Patients with hip fractures, following cardiac surgery, in transition between care settings, long-term care, with multi-morbidities, primary care, acute care, home care, and children with special needs were followed.

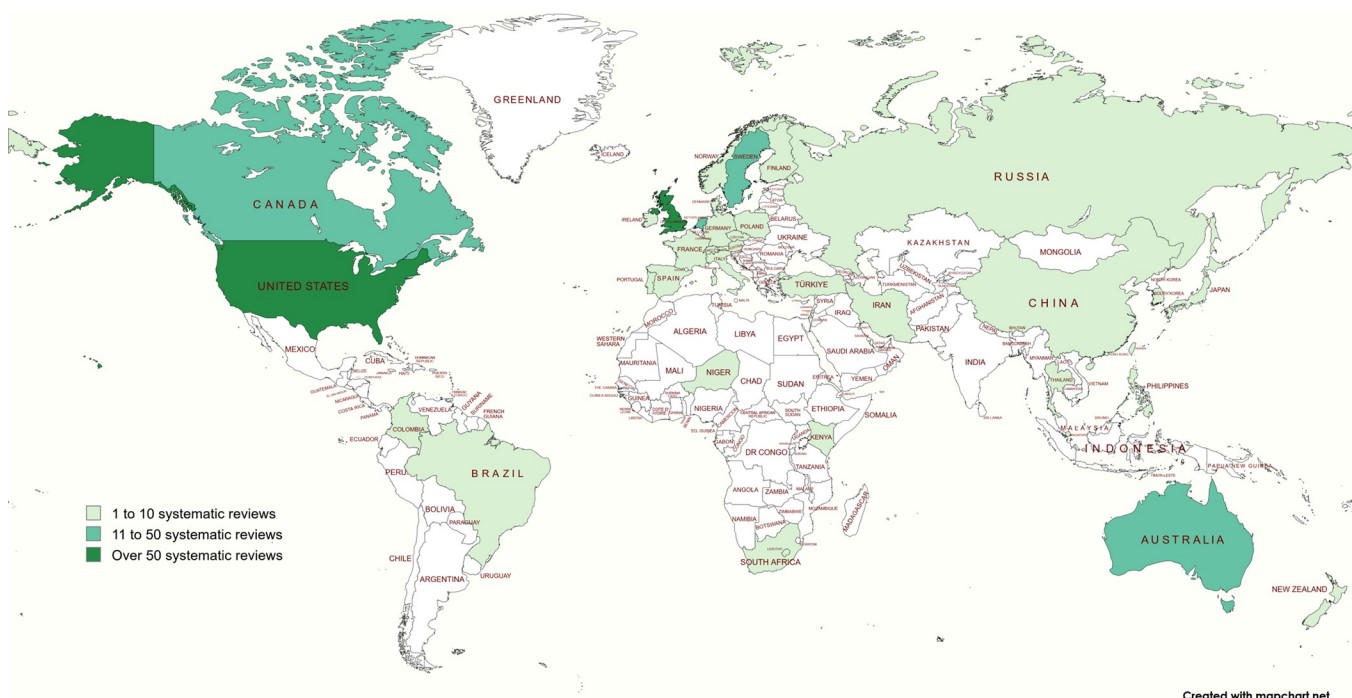

**Fig 2. Number of systematic reviews per country.** We would like to acknowledge the use of MapChart (https://www.mapchart.net) for creating Figs 2 and 3 in this article.

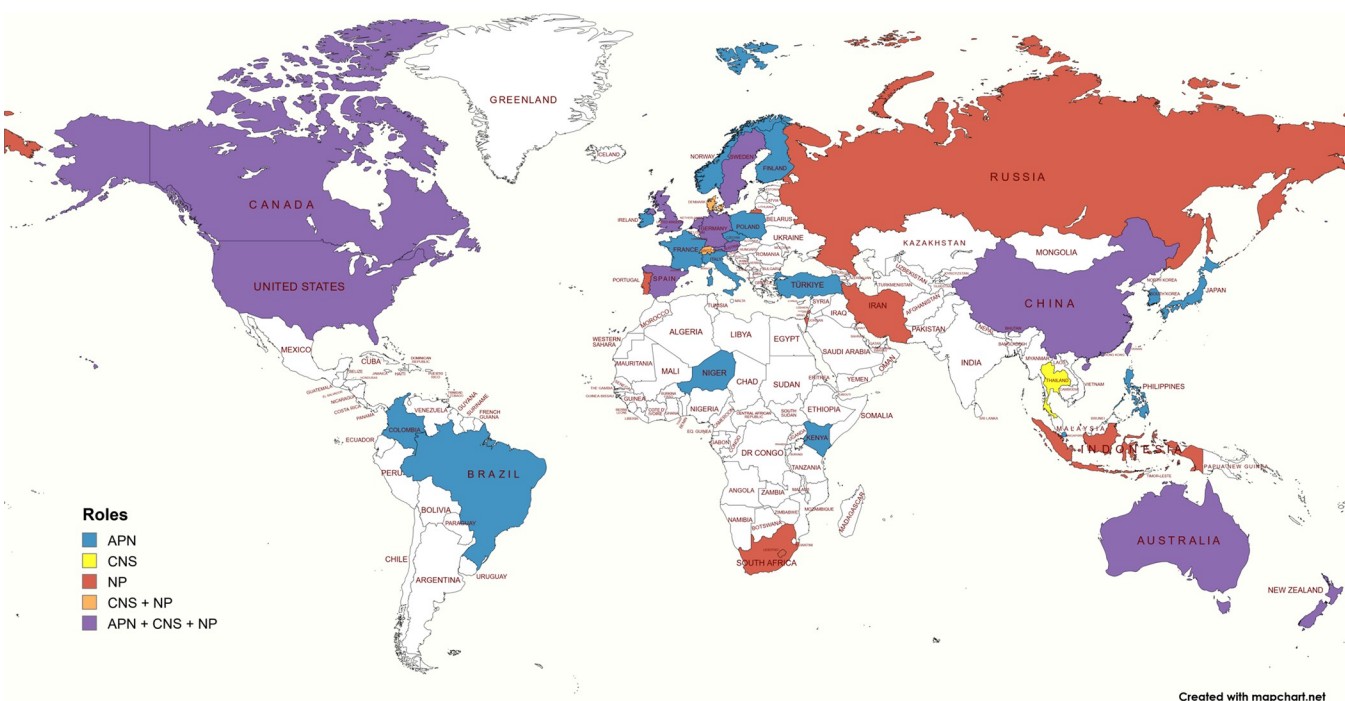

**Fig 3. Advanced practice nursing roles and countries where these roles are represented.** We would like to acknowledge the use of MapChart (https://www.mapchart.net) for creating Figs 2 and 3 in this article.

Equal to statistically significant improvements noted in all reviews. No outcomes favoured the control group.

**Adaptation to Health Conditions** was examined in nine reviews [35, 36, 41–47]. Reviews of APNs in acute and primary care and NPs in primary care examined self-management of health conditions (e.g., osteoporosis, rheumatoid arthritis), lifestyle, self-efficacy, life goal attainment and advance directive for older adults, reductions in disempowerment in home-bound elderly, reductions in uncertainty in women with cancer and depression, reductions in mental distress for adolescents and caregivers. All studies showed equal to statistically significant improvements in the intervention group with no indicators favouring the control group.

**Clinical Outcomes** were divided into health, cardiovascular, cancer care, diabetes, mental health, musculoskeletal, renal and respiratory. Each is presented below.

**Clinical/Health** was assessed in 12 reviews [38–40, 42, 48–56]. Studies included APNs, NPs and CNSs in acute care, and APNs and NPs in primary care. The reviews examined self-reported perceptions of health, physical and mental health, health risk reduction, complex Type 1 and Type 2 diabetes, men with prostate or bladder cancer receiving radiotherapy, and the influence on nurse-led models of care on health condition-specific clinical indicators related to lung function, blood pressure control, lipid profiles, chronic inflammatory arthritis, cancer-related colorectal surgery, human immunodeficiency virus (HIV) and mental illness, chronic depression, women with gestational diabetes, people with chronic kidney disease, nursing observations of sitter days, neonatal complications, mortality or long term delays. All 12 reviews found equal to statistically significant improvements or reduction of harm in favour of the intervention group. No outcomes favoured the control group.

**Clinical/Cardiovascular** outcomes were examined in 18 reviews [7, 34, 39, 40, 47, 50, 57–69] of APNs and NPs in acute and primary care. Care was provided to adults, residents in LTC, and patients 12 weeks post cardiac surgery. Outcomes included assessment and monitoring of patients with heart failure, blood pressure and symptom management, screening for undetected hypertension in the Emergency Room, vascular risk reduction, direct current cardioversion, and metabolic outcomes. Results include statistically significant reductions in systolic and diastolic blood pressure using meta-analyses [50, 60, 64, 65, 67]. Equal to statistically significant reductions noted in outcomes. No outcome favouring the control group was identified.

**Clinical/Cancer Care** was identified in four reviews [70–73]. Studies encompassed APNs and CNSs in acute care and NPs in primary care. Bryant-Lukosius, Cosby, et al. [70] reported that 150 outcomes were measured across 29 studies. Among these, 88 outcomes were equal and 53 were superior in the APN group. Nine outcomes favoured the control group including health perceptions, symptom distress, depression and physical functioning in 1/9 studies [70]. In breast cancer care, statistically significant reductions were noted in levels of uncertainty at one, three and six months but not at 12 months. Statistically significant reductions in uncertainty and symptom distress (p<0.0001), and better health related quality of life (HRQoL) SF-12 related to mental (p = 0.0001) and physical(p<0.001) well-being were noted in the intervention group. Cancer screening was examined in six studies with statistically significant increases noted in the APN group. Similar technical competencies were noted between the intervention and control groups in these studies. NPs had significantly fewer discrepancies between the PAP smears and the biopsies, unsatisfactory colposcopies or missed invasive cancers. CNSs contributed to understanding and meeting the individual needs of women with gynecological cancer in 1/1 study [71]. NPs increased the number of clinical skin examinations by 14% over a 4-year period, and the number of referrals to a dermatologist, biopsies and surgery also decreased during the same period in 1/1 study [72]. CNS-led cancer survivorship and post-treatment follow-up interventions in acute care significantly improved patients' information

needs, self-care, coping and well-being [70]. However, there was a lack of studies examining post treatment survivorship care in primary care [73].

**Clinical/Diabetes** was identified in 11 reviews [34, 38–40, 47, 58, 61, 62, 64–66, 74]. Studies were conducted with APNs in acute and primary care and NPs in primary care. All studies showed equal to statistically significant reductions in HbA1C and blood glucose levels. Significantly more eye and foot exams in the intervention group noted in one review [62]. No differences in annual eye exams noted in another review [66]. No measures favoured the control group.

**Clinical/Mental Health** was examined in 18 reviews [7, 33, 35, 36, 45, 47, 50, 51, 60, 67, 70, 75–81]. Studies included APNs, NPs and CNSs in acute care, and APNs and NPs in primary care. Scheydt [79] mapped 46 tasks and activities of mental health APNs to six practice domains that include 1) clinical nursing, 2) care coordination and case management, 3) psychosocial health promotion and prevention; 4) consulting, education and coaching, 5) leadership and public relations, and 6) research and practice development. Patient health conditions included depression and/or anxiety, anxiety rehabilitation, symptom burden, depression following cardiac surgery, emotional well being, social functioning, psychological morbidity, distress, empowerment, self-management, confidence, subjective health status, and cognitive impairment over 12 months. Equal to statistically significant improvements noted in all the studies. One review identified decreased odds of depression at the 4-year mark for clinic patients in the control group (p = 0.001) [67]. However, the authors cautioned of a suspected cross-over effect.

**Clinical/Musculoskeletal** indicators were identified in six reviews [10, 43, 46, 61, 82, 83]. Reviews included APNs and CNSs in acute care and CNSs and NPs in primary care. Unchanged rates of hip re-fractures in the frail elderly, and disease activity and progression in rheumatoid arthritis were noted. Statistically significant reductions in surgical infections and caregiver distress at 2 weeks post discharge of frail elderly identified in one review of CNSs in transition roles [83]. No outcomes favoured the control group.

**Clinical/Renal** indicators were examined in three reviews [58, 62, 65]. Studies were conducted with NPs in primary care. No significant differences noted in parameters of kidney functioning at six months, phosphate levels, levels of urinary albumin excretion, and urinalysis in the intervention group. No outcomes favoured the control groups.

**Clinical/Respiratory** outcomes were identified in eight reviews [33, 38–41, 62, 65, 84, 85]. Studies were conducted with APNs and NPs in primary care. Equal to statistically significant results favouring the intervention group for the frequency of asthma exacerbation and asthma control in adults and children, airway hyper-reactivity, antibiotic treatments, peak flow measures, and lung function at 12 and 24 months. No outcome favouring the control group was identified.

**Diagnosis** was identified in three reviews [61, 86, 87]. Studies were conducted with NPs in primary care and examined diagnostic accuracy, most common diagnoses, and nurse-initiated X-Rays and treatments. The most common diagnoses in the Emergency room included soft tissue injuries and fractures. Improvements were noted in adenoma detection (p < 0.001), and diagnosis and treatment (p < 0.001) in the intervention group. No outcomes favoured the control groups.

**Education-Patient** was examined in 10 reviews [36, 48, 52, 61, 62, 64, 66, 69, 87, 88]. Studies were conducted with APNs and NPs in acute care and NPs in primary care. Activities included providing information about independent and supplemental nurse prescribing, knowledge of positive contributions of prescribing, health education, medication, symptom relief, discharge information, face-to-face visits and telephone calls at 12 months or longer, providing written documentation, information on who to contact if needed, healthy lifestyle

recommendations for sodium intake, alcohol consumption, and weight management. Equal to statistically significant results favouring the NP group noted for all the reviews. No outcomes in favour of the control group were identified.

**Mortality** identified in 21 reviews [7, 10, 33–36, 39, 40, 46, 61, 64, 67, 80, 84, 89–96] and included APNs and NPs in acute and primary care. Studies examined mortality, mortality at 30-days, 90 days, 12 months and 24 months, mortality related to coronary events, total mortality, all-cause mortality, survival time, and mortality in critical care. Equal to statistically significant reductions in mortality or improved survival noted in all studies. A cross-over effect noted in one study because all patients had the chance to attend the nurse-led clinics within 10 years and detecting differences between the groups was unlikely [67]. One review [10] reported increased mortality at 30 days (p = 0.04) in 1/12 study. No other studies favoured the control group.

**Morbidity** was reported in one review [10]. A trend towards a reduction in the complication rate identified in 6/6 studies, with rates remaining low. One study reported a significant decrease and one study reporting a non-significant increase noted between the intervention and control groups.

**Patient Adherence** was identified in seven reviews [7, 10, 33, 48, 51, 85, 97] and included APNs and NPs in acute and primary care. Indicators included bone mineral density testing, condition-specific medications, adherence to cardiac rehabilitation, dietary management, lipid profiles at 12 months, smoking cessation, patient enablement, follow-up adherence for return appointments. One meta-analysis (n = 676 patients) of medication adherence identified that short-term adherence (pooled odds ratio 1.55 (1.04–2.29) improved over time with long-term adherence (pooled odds ratio: 1.87 (1.35–2.61)) [97]. Equal to statistically significant improvements were noted in the intervention group with no results favouring the control group.

**Quality of Life (QoL)** was examined in 28 reviews [7, 10, 33, 34, 36, 37, 41, 42, 45, 50, 52, 55, 56, 60–62, 66, 67, 75–78, 80, 84, 85, 95, 98, 99]. Studies were conducted with APNs, NPs and CNSs in acute care and APNs and NPs in primary care. Indicators included physical, social and role functioning, health-related QoL, cancer survivorship QoL in acute care, condition-specific QoL for asthma, cancer, diabetes, coronary artery disease, QoL in heart failure at 12 weeks and 12 months, chronic obstructive pulmonary disease (COPD), quality-adjusted life years, apnea and feeding tolerance in neonates, daily functioning and perceived health problems in elderly patients at 6 weeks post hospital discharge. Equal to statistically significant improvements were noted in all reviews. Chan [42] highlighted that 1/17 studies reporting on health related QoL in their review found a small but statistically significant reduction in overall physical component scores on the SF36 for patients with Type 2 diabetes.

**Satisfaction-Patient and Family** was identified in 40 reviews [7, 33, 35, 36, 38–44, 48, 51, 52, 55, 56, 61–63, 69, 71, 76, 78, 80, 87, 88, 92, 94, 96, 100–111]. Studies included APNs, NPs and CNSs in acute care and APNs and NPs in primary care. Almost all reviews found equal to statistically significant improvements in patient and family satisfaction. Two meta-analyses [33, 69] were conducted with significant improvements noted for NP care of patients post hysterectomy and for patient and parent satisfaction in primary care. Mixed findings noted in two reviews. The first review [55] found that men with prostate cancer had statistically lower scores for satisfaction at the end of treatment in ½ studies. The second review [108] highlighted that patient satisfaction with the usual source of care and wait times was worse in states with the least restrictive NP scope of practice policies.

**Signs and Symptoms** were examined in 18 reviews [33, 36, 42, 46, 56, 60, 61, 66, 67, 75, 77, 78, 85, 92, 99, 110, 112, 113]. Studies included APNs in acute and primary care and NPs in primary care. Outcomes included symptom burden, symptom management for angina, appetite loss, constipation, diarrhea, dyspnea, insomnia, nausea and vomiting, cognitive and

behavioural changes and dementia at 12 months and 18 months, fatigue at 12 months and 24 months, urinary tract infections, symptom improvement, pain, jaundice in neonates. Equal to statistically significant improvements in signs and symptom management noted in the reviews. In a meta-analysis of appetite loss conducted by Monterosso (2019) [60], no significant differences were noted in appetite loss at the end of the intervention (up to 24 months) but appetite loss at 4–6 months was significantly lower in the control group (MD = 4.43, 95%CI [0.08, 8.78], 354 participants, p = 0.05; I2 = 0%).

## Provider indicators

**Adherence to Best Practice-Provider** was identified in 10 reviews [33, 34, 48, 51, 66, 67, 81, 92, 114, 115]. Studies included APNs in acute and primary care and NPs in primary care. Reviews examined medication adherence, renin-angiotensin converting enzyme inhibitors, beta blockers, aspirin intake, clopidogrel, annual physical assessment, condition-specific care, meeting clinical targets, motivational interviewing, dementia care, counselling, cognitive behavioural and problem-solving therapy in older adults. Equal to statistically significant improvements noted in all reviews. No negative outcomes noted.

**Education-Provider** was examined in 17 reviews [34, 44, 59, 72, 87, 92, 107, 114–123]. Studies were conducted with APNs and NPs in education and acute and primary care. Provider education topics included pharmacology, clinical decision-making, task knowledge and capabilities, self-knowledge and interpersonal skills, transition as new faculty, advanced diagnostic procedures and skills, educational preparedness in the Emergency Room, mentor competency, informal training to assess skin lesions, clinical skin assessments, early detection of skin cancer, application of evidence-based protocols and staff consultation. Training modalities used were didactic, e-learning, workshops, education logs, simulation, face-to-face and observations by experts, orientation in LTC and in academia, continuing education and on-the-job training. Equal to statistically significant improvements and no negative outcomes noted across the reviews.

**Illness Prevention** was reported in nine reviews [33, 45, 46, 48, 61, 62, 67, 70, 73]. Care involved APNs in acute and primary care and NPs in primary care. Follow-up consultations, metabolic outcomes, feasibility of screening for depression, patient health assessment at 12 and 24 months, clinical examinations, smoking status, smoking cessation, diet at one-year, cervical cancer screening, breast cancer screening, colorectal cancer screening, follow-up consultations and return visits were examined. No cancer screening activities for patients were identified for APNs in acute care in one review [70]. Equal to statistically significant improvements noted.

**Interprofessional Team Functioning** was examined in five reviews [34, 35, 81, 88, 117]. Studies include APN in acute care and APNs and NPs in primary care. Reviews reported on communication, building trusting relationships, reducing hierarchies, interdisciplinary approaches, team functioning, and collaborative care. Equal to improved team functioning noted across reviews. No provider outcomes reported in LTC [34].

**Prescribing** identified in 24 reviews [34, 36, 42, 46, 48, 51, 61, 62, 69, 81, 85, 88–90, 92, 112–115, 117, 124–127]. APNs, NPs and CNSs in acute care and APNs and NPs in primary care were studied. A wide range of prescribing activities were described including medications, systemic anti-cancer therapy, blood tests, laboratory tests, weight loss pharmaceuticals, rescue medication, opioid prescribing. Equal to statistically significant results favouring the intervention group noted in 23/24 reviews. One review [112] found that NPs appeared to be less likely to prescribe an opioid for acute users at baseline but they were more likely to prescribe higher doses of morphine milligram equivalents in 2/2 studies. Single studies in reviews by

Salamanca-Balen [126] in acute care and Swan [51] in primary care found that CNSs and APNs, respectively, requested significantly more tests.

**Satisfaction-Provider** was examined in nine reviews [42, 92, 105, 114, 115, 118, 128–130] and included APNs and NPs in acute and primary care. Positive views were noted with improved communication, good interpersonal relationships, increased role confidence, improved dementia care, care coordination and timely access to care. Some mental health nurse prescribers expressed anxiety over role conflict related to prescribing in one review [114]. Dissatisfaction was associated with higher overtime hours in one review [128].

## Health system indicators

**Access to Care** was identified in eight reviews [49, 54, 61, 71, 100, 115, 121, 126]. Studies included CNSs in acute and primary care and APNs and NPs in primary care. Studies examined access for women with gynecological cancers, treatment for precancerous lesions, patients with COPD, primary care, and pediatric visits. Equal to statistically significant improvements noted in access to care. No negative outcomes noted.

**Consultations** were reported in 20 reviews [7, 33, 34, 42, 46, 48, 51, 53–55, 61, 62, 69, 72, 85, 92, 105, 109, 122, 126]. Studies included all roles but did not report on CNSs in primary care separately. Mixed findings noted with equal to statistically significant reductions in consultations, treatment initiation in lung cancer, and improved attendance to provider appointments in acute and primary care noted in 10 reviews [7, 33, 42, 51, 72, 85, 92, 105, 122, 126]. However, one study in the Salamanca-Balen [126] review reported that the number of referrals to primary care and cardiology clinics increased significantly. Increased consultations favouring the control group noted in nine reviews [34, 46, 48, 53–55, 62, 69, 109]. One review did not report on the length of consultations times [61].

**Costs** were reported in 40 reviews [7, 10, 33–36, 38–40, 42, 44, 46, 47, 49, 51, 52, 54, 55, 57, 60, 61, 63, 69, 83–85, 91–96, 98, 103, 106, 109, 111, 115, 126, 131, 132]. Studies included all roles. Costs were estimated for diverse goods and services including laboratory tests, diagnostic procedures, direct and indirect costs, medications, healthcare personnel, patient care visits, types of services, salary, fee for service, hospital costs, home care, program costs, quality adjusted life years, cost effectiveness, incremental cost effectiveness ratio, cost utility, and total costs. Equal to reduced costs noted in 32 reviews. Mixed results noted in seven reviews [42, 52, 55, 84, 109, 126, 131] at the level of individual studies included in the reviews. One review [63] did not report costs in retained studies of APNs.

**Emergency Room Visits** were identified in 18 reviews [34, 37–40, 45, 47, 53, 62, 69, 84, 86, 90, 93, 102–104, 126, 133]. Except for one review conducted with CNSs in acute care [126], reviews were conducted with APNs and NPs in primary care. Equal to statistically significant reductions were noted in 15 studies. Mixed results noted in two reviews [34, 45] with equal to statistically significant increases noted at the study level. Lawton [84] did not specifically report on ER attendance. Van Vliet [53] did not report on non conveyance rates for NPs as data were not available for ambulance transport.

**Health Care Service Delivery** was reported in 25 reviews [33, 34, 36, 37, 41, 45, 46, 51–56, 61, 62, 69, 78, 83, 92, 93, 99, 100, 103, 122, 127]. Studies conducted with APNs and NPs in acute and primary care and CNSs in acute care examined adherence to appointments, hospitalizations, office-based visits to oncologists, no-show rates, continuity of care of providers, transitional care for patients with schizophrenia, home-based interventions for individuals with severe mental illness and HIV, primary care interventions for women with ovarian cancers, healthcare utilization, follow-up contact, barriers to pre-exposure prophylaxis implementation in HIV, acceptability of nurse-led care, specialist and primary care visits, collaborative

care models, end of life care for patients with severe dementia, palliative care, number of outpatient contacts and primary care contacts, service use at 30-, 120- and 180-days, transition care, case management consultation length, NP growth over time, post-discharge care, and return visits. Lovink [34] reported that no study measured implementation as a process outcome. Equal to significant reductions in healthcare service delivery or resource utilization noted in 19 reviews. Yang [54] noted that full practice authority for NPs led to increases in service utilization. Mixed findings noted in four reviews [51, 55, 56, 69]. These reviews respectively reported an increased number of visits at 12 months but fewer APN visits at 24 months in primary care, more referrals to mental health specialists and prenatal visits, more home care services for elderly patients post discharge, and additional return visits within 2 weeks. Zhang [127] identified key barriers to implement pre-exposure prophylaxis in HIV. Trend to decreased Emergency medical service use for patients with high 911-call use and no significant differences in NP follow up contact after the completion of pre-hospital care [53].

**Hospitalization** was examined in 29 reviews [7, 10, 33, 34, 36–40, 47, 49, 50, 55, 57, 61, 62, 67, 69, 83–85, 89–91, 93, 95, 98, 115, 126, 133]. Reviews included APNs in acute and primary care, NPs in primary care and CNSs in acute care. Reviews examined a range of provider interventions, hospital admissions, heart failure readmissions, re-hospitalizations at 3-, 6- and 12 months, unplanned hospital transfers, ICU readmissions, return visits for any reason, post-discharge care following surgery and for women with high-risk pregnancies and in the post-partum period, nursing home admissions, and home visits. Equal to statistically significant reductions noted in 26 reviews. Mixed results noted in one review where ¼ studies reported an increase in admissions at 2 weeks post-discharge [10]. A higher proportion of hospital admissions [84] and increased admission rates for patients with lung disease reported in 1/1 study [61].

**Length of Stay** was reported in 22 reviews [7, 10, 36, 39, 40, 42, 50, 55, 56, 62, 63, 83, 86, 87, 91, 93, 94, 96, 104, 109, 115, 126, 134]. Studies with APNs, NPs and CNSs in acute care and APNs and NPs in primary care examined length of stay in the intensive care unit, in the Emergency room, in radiology, length of visits, in hospital, psychiatric inpatient unit, post-partum and postoperative cardiac surgery. Equal to statistically significant reductions noted 19 reviews. Mixed results noted in three reviews [62, 115, 126].

**Patient Safety** was examined in 19 reviews [7, 33, 35, 36, 39, 40, 46, 54–56, 61–63, 72, 87, 90, 91, 103, 109, 135]. Studies included APNs, NPs and CNSs in acute care and NPs in primary care. Safety issues included postoperative complications, urinary tract infections, time to occlusion of arteriovenous fistula access, adverse events, treatment complications, medication interactions, malpractice, restraint use, sitter walk-aways, falls, mandatory monitoring of blood tests, out of range blood tests, risk factor management, missed injuries, deep vein thrombosis identification, inappropriate management of health conditions, unnecessary biopsies, condition-specific complications related to strokes, pregnancy, and the intensive care unit. Equal to statistically significant reductions noted for all adverse events. No studies favoured the control groups.

**Quality of Care** reported in 11 reviews [34, 42, 49, 52, 54, 85, 92, 103, 106, 135, 136]. Studies included APNs and CNSs in acute care and APNs and NPs in primary care. Indicators included quality of outreach services for dementia care, complications related to endoscopies, improved recording of medical information, incident rates of arteriovenous fistula use, incidence of central venous catheter placement at the start of dialysis, prevalence of late referrals of arteriovenous fistulas, emergency care, and asthma care. Equal to statistically significant improvements noted across the reviews. No outcomes favouring the control groups were noted.

**Scope of Practice** was assessed in 22 reviews [49, 54, 86, 87, 92, 101, 107, 108, 117, 122, 130, 136–146]. Several reviews used meta-synthesis and thematic analysis to summarize key findings for this indicator. Scope of practice includes clinical and non-clinical role dimensions [145, 146]. Legislation shapes practice [139]. Leadership, consultancy, networking and the ability to develop relationships influence scope of practice [145, 146]. Central themes to changes in scope of practice included professional boundaries, autonomy, interprofessional collaboration and the advanced practice role as a resource for the team [137]. APN governance and regulations are often country-specific [136]. Several reviews indicated that autonomous or full scope of practice enhanced patient, provider and health system level outcomes (e.g., Dawson, 2015; Hutchison, 2014) [101, 141].

**Wait Times** were examined in 12 reviews [10, 53, 61, 63, 72, 86, 96, 100, 103, 104, 111, 122]. Studies included APNs and NPs in acute care and primary care. Wait times for treatment, for appointments, in the Emergency Room, did-not-wait rates, time to surgery, to provider's initial assessment, patients seen in a timely manner, and on scene treatment times were measured. Equal to statistically significant reductions noted in all reviews. No data for APN wait times for direct current cardioversion [63] or pre-hospital conveyance [53].

**Artificial Intelligence-Health Technology** was examined in two reviews [147, 148]. One review identified 13 barriers to hospital-based NPs using clinical decision support [147]. A second review examined NP involvement and experience with AI-based health technologies in primary care (e.g., family, geriatric and pediatric care), hospital (e.g., acute, post-acute and post-operative), and in the Emergency Room in which Machine Learning-based clinical decision support systems were developed. Decision outputs (diagnostic and referral decisions) were compared with the clinical decisions of NPs who assumed the role of diagnostic and/or therapeutic expert. NP clinical activities that are enabled by AI-based health technologies (AIHT) systems and addressed AIHT in support of the referral or triage decisions made by NPs were examined. Studies examined AI-based technologies were meant to improve the follow-up and surveillance of patients by NPs, and a single study in the review examined illness prevention.

## Discussion

We sought to determine if current systematic reviews represent countries where APNs, NPs or CNSs are found, and describe the studies, study population, roles, and outcomes identified in the reviews. The overview identified 117 systematic reviews published in 121 papers incorporating 1653 primary studies. Less than 4% of papers were cited more than twice, thus there was minimal overlap in our findings. An average of 3.4 countries per review were identified. Countries (e.g., United States) with longer histories of advanced practice nursing roles represented the largest contributors to the overview. Countries where APN, NP and CNS roles are emerging were also identified (e.g., France) [149–151]. Geographical gaps in advanced practice nursing research were identified in countries in almost all WHO regions including Africa, Europe, Southeast Asia, Eastern Mediterranean, Western Pacific and Latin America.

The evaluation of the risk of bias was conducted using CASP and several criteria were met. However, the assessments related to the identification of all important studies was low. Although several databases were searched, review authors did not clearly state that two reviewers had independently screened titles and abstracts and extracted the relevant studies. If was often not clear if a second reviewer had verified the extractions. This underscores the importance of adhering to reporting guidelines (e.g., PRISMA 2020 [18]) when reporting on this important step of the review. In addition, reviews were also downgraded if they included geographical and language restrictions.

We found highly consistent evidence about the care provided by APNs, NPs and CNSs across 29 indicator categories at the patient, provider and health system levels. Care was assessed as equal or superior to the comparator (usual care, physicians) across a wide range of clinical settings, patient populations and acuity levels in almost all the reviews. Mixed findings were noted for quality of life, consultations, costs, emergency room visits, and health care service delivery where some studies favoured the control groups. However, no review or indicator category clearly favoured the control group.

Some areas have been examined extensively as evidenced by the number of reviews included in the indicator category. Indicators related to clinical categories (health, cardiovascular, diabetes, mental health), education-patient, mortality, quality of life, satisfaction-patient and family, signs and symptoms, adherence to best practice-provider, education-provider, prescribing, consultations, costs, emergency room visits, health care service delivery, hospitalization, length of stay, patient safety, quality of care, scope of practice and wait times were examined in 10 to 40 systematic reviews.

The current overview built on the categorization scheme used in a recent overview of indicators to measure the quality of primary healthcare NP practice where outcomes were classified in 26 indicator categories [9]. Three new indicator categories were created for the current overview that incorporated a broader range of advanced practice nursing roles across acute and primary care settings. At the patient level, an indicator related to morbidity was created. At the provider level, satisfaction was added. Artificial intelligence-Health technology was added as a cross-cutting indicator category because of the broad scope and reach of AI. Finally, the indicator category related to rheumatoid arthritis was broadened to include musculoskeletal outcomes. This categorization scheme can be used in subsequent studies to classify and measure relevant APN, NP and CNS outcomes across sectors and with a wide range of patient populations.

There is emerging research in AI that cuts across care sectors by engaging remote monitoring, improving diagnosis accuracy, assisting clinical decision and facilitating predictive analysis. The COVID-19 pandemic increased the use of technology exponentially. Healthcare providers need to have the most up-to-date knowledge to adapt to a quickly changing healthcare landscape. In addition, patients in vulnerable situations (e.g., homelessness) or those with cognitive or sensory deficits (e.g., patients with impaired hearing, patients with dementia) have specific needs regarding technology. Subsequent research needs to close the knowledge gap related to the use of technology and how they influence patient, provider, and health system outcomes. In addition, the use of big data to capture advanced practice nursing contributions is needed. Many health systems worldwide do not track advanced practice nursing care providers adequately. Thus, it is difficult to identify their activities and associated outcomes.

There is a lack of reviews related to APNs, NPs or CNSs and global health research, such as disaster, communicable diseases, and care equity. Other gaps in research in clinical settings include pre-hospital and ambulance care, cancer survivorship in primary care, care of patients in vulnerable situations, interprofessional team functioning, workload and working conditions, acute care nephrology and hemodialysis. In relation to advanced practice roles, there is a lack of research describing CNS roles in primary care and with specific patient populations (e.g., CNSs in adolescent care).

Nurses in advanced practice roles play an important role in engaging patients and families as partners in care. Patient and family engagement in care needs to be better understood to support their experience of care and engagement (e.g., patient reported experience measures (PREMs) and patient reported outcome measures (PROMs). Other important knowledge gaps include an examination of spirituality, cultural safety and cultural sensitivity and advanced practice nursing [9, 152]. Turkelson et al. [153] highlighted that simulation with NP students

increased cultural sensitivity with patients of Hispanic origins. The umbrella review to measure the quality of primary healthcare NP practice [9] found no indicator to measure cultural safety, particularly with Indigenous Peoples. Filling these gaps in knowledge is critical to give patients and families a voice in their healthcare [154].

Current knowledge about the influence of high and low fidelity simulation on provider behaviours and patient, provider and system outcomes remains mostly anecdotal [155]. We identified two reviews examining simulation with NPs in acute and primary care that focussed on increasing knowledge and learner satisfaction [118, 123]. A systematic review of short team interventions [25] identified helpful modalities for high and low fidelity simulations and debriefing for providers. However, new knowledge is needed to outline content areas for specific patient (e.g., complex care, home care), and provider (e.g., complex technical skills) groups including other advanced practice nursing roles.

## Strengths and limitations

Several challenges were identified during title and abstract screening, extraction and analysis for studies that used the title of nurse-led. Internationally, some countries do not have recognized APN, NP or CNS titles which may have limited our ability to identify this body of research. To counteract this, we included several strategies to optimize study identification including a wide database search, review of the grey literature, hand searches and consultations across our global research collaborators to identify published and unpublished reviews that met our criteria. In addition, we went beyond accepted role titles to review role descriptions in all the reviews that included the term nurse-led. Several reviews were excluded late in the review process, after full text review and extraction, to ensure that we captured roles that met the level of decision-making autonomy consistent with advanced practice nursing roles.

## Conclusion

Our overview identified 117 systematic reviews examining advanced practice nursing across the globe with 38 countries represented across the reviews. Review quality was downgraded because study selection and extraction were not conducted by two reviewers and language or geographic restrictions were applied. However, given the large number of primary studies that were captured, we do not anticipate that this influences our findings. Consistent evidence of equal to statistically significant improvements noted for APNs, NPs and CNSs across 29 indicator categories at the patient, provider and health system levels. Care was assessed across a range of clinical settings, patient populations and acuity levels. Mixed results noted for the length of consultations and costs primarily. No indicator clearly favoured the control group. There is emerging research in AI that cuts across care sectors. Specific research gaps were identified to guide subsequent needs for advanced practice research. They include research related to AI, technology, global health research, interprofessional team functioning, workload and working conditions, how to engage patients and families as partners in healthcare, and care of patients in vulnerable situations.

## Supporting information

**S1 Checklist. PRISMA 2020 checklist.**
(PDF)

**S1 Appendix. Record of review-related decisions.**
(PDF)

**S2 Appendix. Search strategies for the published literature.**
(PDF)

**S3 Appendix. Search strategies for the grey literature.**
(PDF)

**S1 Table. Risk of bias assessment for the included systematic reviews (n = 117).**
(PDF)

**S2 Table. Description of included systematic reviews (n = 117).**
(PDF)

**S3 Table. Extractions of review results by indicator category at the Patient level.**
(PDF)

**S4 Table. Extraction of review results by indicator category at the Provider level.**
(PDF)

**S5 Table. Extraction of review results by indicator category at the Health System level.**
(PDF)

**S6 Table. Extraction of review results by indicator category related to Artificial Intelligence.**
(PDF)

## Acknowledgments

The collaborative effort of the International Council of Nurses Nurse Practitioner Advanced Practice Nurse Network (ICN NP/APNN) is exemplified in this work, with significant contributions made by several ICN NP/APNN members.

## Author Contributions

**Conceptualization:** Kelley Kilpatrick, Isabelle Savard, Li-Anne Audet, Renée Atallah, Mira Jabbour, Wentao Zhou, Kathy Wheeler, Elissa Ladd, Deborah C. Gray, Colette Henderson, Lori A. Spies, Heather McGrath, Melanie Rogers.

**Data curation:** Kelley Kilpatrick, Isabelle Savard, Li-Anne Audet, Gina Costanzo, Mariam Khan, Renée Atallah, Mira Jabbour, Wentao Zhou, Kathy Wheeler, Elissa Ladd, Deborah C. Gray, Colette Henderson, Lori A. Spies, Heather McGrath, Melanie Rogers.

**Formal analysis:** Kelley Kilpatrick, Isabelle Savard, Li-Anne Audet, Gina Costanzo, Mariam Khan, Renée Atallah, Mira Jabbour, Wentao Zhou, Kathy Wheeler, Elissa Ladd, Deborah C. Gray, Colette Henderson, Lori A. Spies, Heather McGrath, Melanie Rogers.

**Funding acquisition:** Kelley Kilpatrick.

**Investigation:** Kelley Kilpatrick, Isabelle Savard, Li-Anne Audet, Gina Costanzo, Mariam Khan, Renée Atallah, Mira Jabbour, Wentao Zhou, Kathy Wheeler, Elissa Ladd, Deborah C. Gray, Colette Henderson, Lori A. Spies, Heather McGrath, Melanie Rogers.

**Methodology:** Kelley Kilpatrick, Isabelle Savard, Li-Anne Audet, Renée Atallah, Mira Jabbour, Wentao Zhou, Kathy Wheeler, Elissa Ladd, Deborah C. Gray, Colette Henderson, Lori A. Spies, Heather McGrath, Melanie Rogers.

**Project administration:** Kelley Kilpatrick, Renée Atallah.

**Software:** Renée Atallah, Mira Jabbour.

**Supervision:** Kelley Kilpatrick.

**Validation:** Kelley Kilpatrick, Isabelle Savard, Li-Anne Audet, Gina Costanzo, Mariam Khan, Renée Atallah, Mira Jabbour, Wentao Zhou, Kathy Wheeler, Elissa Ladd, Deborah C. Gray, Colette Henderson, Lori A. Spies, Heather McGrath, Melanie Rogers.

**Writing – original draft:** Kelley Kilpatrick, Isabelle Savard, Li-Anne Audet, Gina Costanzo, Renée Atallah, Mira Jabbour, Wentao Zhou, Kathy Wheeler, Elissa Ladd, Deborah C. Gray, Colette Henderson, Lori A. Spies, Heather McGrath, Melanie Rogers.

**Writing – review & editing:** Kelley Kilpatrick, Isabelle Savard, Li-Anne Audet, Gina Costanzo, Mariam Khan, Renée Atallah, Mira Jabbour, Wentao Zhou, Kathy Wheeler, Elissa Ladd, Deborah C. Gray, Colette Henderson, Lori A. Spies, Heather McGrath, Melanie Rogers.

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
