## [Decision Letter · Decision Letter 0]

20 Mar 2024

PONE-D-23-42783A global perspective of advanced practice nursing research: A review of systematic reviewsPLOS ONE

Dear Dr. Kilpatrick,

Thank you for submitting your manuscript to PLOS ONE. After careful consideration, we feel that it has merit but does not fully meet PLOS ONE’s publication criteria as it currently stands. Therefore, we invite you to submit a revised version of the manuscript that addresses the minor points raised during the review process.

We look forward to receiving your revised manuscript.

Kind regards,

Federica Canzan

Academic Editor

PLOS ONE

2. We note that Figures 1 and 2 in your submission contain [map/satellite] images which may be copyrighted. All PLOS content is published under the Creative Commons Attribution License (CC BY 4.0), which means that the manuscript, images, and Supporting Information files will be freely available online, and any third party is permitted to access, download, copy, distribute, and use these materials in any way, even commercially, with proper attribution. For these reasons, we cannot publish previously copyrighted maps or satellite images created using proprietary data, such as Google software (Google Maps, Street View, and Earth). For more information, see our copyright guidelines: http://journals.plos.org/plosone/s/licenses-and-copyright.

a. You may seek permission from the original copyright holder of Figures 1 and 2 to publish the content specifically under the CC BY 4.0 license. 

Reviewers' comments:

Reviewer's Responses to Questions

**Comments to the Author**

1. Is the manuscript technically sound, and do the data support the conclusions?

Reviewer #1: Yes

Reviewer #2: Yes

2. Has the statistical analysis been performed appropriately and rigorously? 

Reviewer #1: N/A

Reviewer #2: Yes

3. Have the authors made all data underlying the findings in their manuscript fully available?

Reviewer #1: Yes

Reviewer #2: Yes

4. Is the manuscript presented in an intelligible fashion and written in standard English?

Reviewer #1: Yes

Reviewer #2: Yes

5. Review Comments to the Author

Reviewer #1: Congratulations to the team of the International Council of Nurses Nurse Practitioner Advanced Practice Nurse Network that performed this huge umbrella review. Despite the amount of studies included, it is clearly presented, both the methodological aspects of the revision and the results. Methods are presented in detail as expected in a systematic review, and results are synthetized in broader categories, previously developed, allowing the reader to have a grasp of the types of outcomes of APN, NP and CNS roles.

I would suggest minor improvements, namely:

i) Abstract

I suggest that under results the sentence written in lines 652-657 expresses more clearly the results of the study "We found highly consistent evidence about the care provided by APNs, NPs and CNSs across 29 indicator categories at the patient, provider and health system levels. Care was assessed as equal or superior to the comparator (usual care, physicians) across a wide range of clinical settings, patient populations and acuity levels in almost all the reviews. Mixed findings were noted for quality of life, consultations, costs, emergency room visits and health care service delivery where some studies favoured the control groups."

ii) Matherials and Methods:

In Database search (line 186) it is not clear why you decide to include reviews published after 2011.

In the assessment of review quality it is not clear if the authors decided to include the reviews even the ones with low CASP scores.

iii) Results

In line 291 authors refer to Patient outcomes and in line 301 Patient Indicators. Are they synonyms?

Reviewer #2: I thank the authors for giving me the opportunity to read their manuscript which deals with an important topic in nursing. In the PRISMA statement, enter how many articles were found for each database queried.

I would also insert a table with search strings for each database queried.

6. PLOS authors have the option to publish the peer review history of their article (what does this mean?). If published, this will include your full peer review and any attached files.

Reviewer #1: No

Reviewer #2: No

---

## [Author Response · Author response to Decision Letter 0]

28 Mar 2024

Please see enclosed Response to Reviewers for our response to their comments.

---

## [Decision Letter · Decision Letter 1]

22 May 2024

A global perspective of advanced practice nursing research: A review of systematic reviews

PONE-D-23-42783R1

Dear Dr. Kilpatrick,

We’re pleased to inform you that your manuscript has been judged scientifically suitable for publication and will be formally accepted for publication once it meets all outstanding technical requirements.

Kind regards,

Federica Canzan

Academic Editor

PLOS ONE

Additional Editor Comments (optional):

Reviewers' comments:

Reviewer's Responses to Questions

**Comments to the Author**

1. If the authors have adequately addressed your comments raised in a previous round of review and you feel that this manuscript is now acceptable for publication, you may indicate that here to bypass the “Comments to the Author” section, enter your conflict of interest statement in the “Confidential to Editor” section, and submit your "Accept" recommendation.

Reviewer #1: All comments have been addressed

2. Is the manuscript technically sound, and do the data support the conclusions?

Reviewer #1: Yes

3. Has the statistical analysis been performed appropriately and rigorously? 

Reviewer #1: N/A

4. Have the authors made all data underlying the findings in their manuscript fully available?

Reviewer #1: Yes

5. Is the manuscript presented in an intelligible fashion and written in standard English?

Reviewer #1: Yes

6. Review Comments to the Author

Reviewer #1: The manuscript required minor revisions and the authors accepted and changed all the suggestions. Congratulations to the team for this detailed and rigorous systematic review.

7. PLOS authors have the option to publish the peer review history of their article (what does this mean?). If published, this will include your full peer review and any attached files.

Reviewer #1: No

---

## [Editor Report · Acceptance letter]

27 May 2024

PONE-D-23-42783R1 

PLOS ONE

Dear Dr. Kilpatrick, 

I'm pleased to inform you that your manuscript has been deemed suitable for publication in PLOS ONE. Congratulations! Your manuscript is now being handed over to our production team.

Kind regards, 

on behalf of

Professor Federica Canzan 

Academic Editor

PLOS ONE